# Beyond Cellular Immunity: On the Biological Significance of Insect Hemocytes

**DOI:** 10.3390/cells12040599

**Published:** 2023-02-12

**Authors:** David Stanley, Eric Haas, Yonggyun Kim

**Affiliations:** 1Biological Control of Insect Research Laboratory, USDA/ARS, 1503 S Providence Road, Columbia, MO 65203, USA; 2Department of Chemistry and Biochemistry, Creighton University, 2500 California Plaza, Omaha, NE 68178, USA; 3Department of Plant Medicals, Andong National University, Andong 36729, Republic of Korea

**Keywords:** hemocyte, immunity, molting, development, reproduction, apoptosis, cancer

## Abstract

Insect immunity is assorted into humoral and cellular immune reactions. Humoral reactions involve the regulated production of anti-microbial peptides, which directly kill microbial invaders at the membrane and intracellular levels. In cellular immune reactions, millions of hemocytes are mobilized to sites of infection and replaced by hematopoiesis at a high biological cost after the immune defense. Here, we considered that the high biological costs of maintaining and replacing hemocytes would be a better investment if hemocytes carried out meaningful biological actions unrelated to cellular immunity. This idea allows us to treat a set of 10 hemocyte actions that are not directly involved in immunity, some of which, so far, are known only in *Drosophila melanogaster*. These include (1) their actions in molting and development, (2) in surviving severe hypoxia, (3) producing phenoloxidase precursor and its actions beyond immunity, (4) producing vitellogenin in a leafhopper, (5) recognition and responses to cancer in *Drosophila*, (6) non-immune actions in *Drosophila*, (7) clearing apoptotic cells during development of the central nervous system, (8) developing hematopoietic niches in *Drosophila*, (9) synthesis and transport of a lipoprotein, and (10) hemocyte roles in iron transport. We propose that the biological significance of hemocytes extends considerably beyond immunity.

## 1. Introduction

Insect innate immunity is resolved into two major sections: humoral and cellular reactions to infection, invasion, and wounds. Humoral immunity entails the regulated expression of genes encoding anti-microbial peptides (AMPs) and the conversion of prophenoloxidase (PPO) into active phenoloxidase (PO). AMPs appear in the hemolymph of infected insects approximately 6–12 h post infection (hpi). AMPs directly kill infecting microbes by disrupting their membranes and interfering with internal mechanisms. AMPs are not limited to insects. Bacteria, archaea, protozoans, fungi, plants, and animals produce thousands of AMPs [1]. Active PO acts in wound closure and repair, and it acts in immunity.

Cellular immunity includes the mobilization of hemocytes to sites of infection and wounds. Hemocytes directly interact with invading microbial cells, phagocytosing or killing them at the membrane or intracellular levels. Cellular immune responses are launched immediately after an infection is recognized and they significantly reduce numbers of invading microbes within the first hpi in *Manduca sexta* [2]. Many bacterial cells are cleared from circulation by formation of melanized nodules. In this process, individual hemocytes form microaggregates of a few hemocytes with attached dead and dying microbes. These microaggregates grow into nodules by accumulating more microbe-laden hemocytes, and the nodules are finally melanized by active PO and attached to the surfaces of internal organs and inner body walls. In some insect species, the nodules can be routinely counted as a quantitative measure of an immune reaction. In *M. sexta*, cellular immune responses to microbial infections are biologically expensive because millions of hemocytes are invested in the nodulation process following a single microbial infection [3]. These millions of lost cells are replaced within hemopoietic organs. Fifth-instar *M. sexta* larvae are large insects, and smaller insects invest less than millions of hemocytes into cellular immune reactions.

Insect immunity has been reviewed many times from a range of perspectives. Here, we note that *Drosophila melanogaster* hemocytes and their roles in immunity have been studied for decades ([4] and his many other publications on the topic). Gillespie et al. [5] provided a cogent basis for grasping a broad view of immunity. Lemaitre and Hoffman [6] focused on *Drosophila* immunity to infections, and later, Yang et al. [7] treated *Drosophila* cellular and humoral immunity to parasitoids. Lazzaro [8] served as guest editor of a special issue of Insects with 11 papers devoted to insect immunity. Kingsolver and Hardy [9] reviewed intracellular immune signaling pathways. Cooper and Eleftherianos [10] discussed the possibility that insects express some forms of immune priming and memory. Kim and Stanley treated eicosanoid signaling mechanisms in insect immunity [11,12,13]. We note that this brief list is not exhaustive.

Moret and Schmid-Hempel [14] addressed the cost of immunity. The authors injected bumblebee workers, *Bombus terrestris*, with lipopolysaccharide (LPS), which stimulates immune reactions in a non-pathological manner [15]. Their data show that challenging worker bumblebees with LPS led to very high worker deaths within 25–30 hpi, which they interpreted in terms of immunity costs. In addition, Zhang et al. [16] reported that prostaglandin signaling mediates immune recovery from bacterial, *Serratia marcescens*, infections in larvae of the fall armyworm, *Spodoptera frugiperda*. The recovery was attended by developmental delays and early deaths, again, documenting the biological costs of immunity. Ardia et al. [17] directly measured increased metabolic rates following an immune challenge. The authors reported that inducing an encapsulation reaction led to as much as a 28% increase in resting metabolic rates in *Acheta domesticus*, *Periplaneta americana,* and *Tenebrio molitor*, determined as increased production of CO_2_. Replacing very large numbers of hemocytes is part of that high cost. We hypothesized that the cost of maintaining and replacing a standing army of circulating and sessile hemocytes would be a better biological investment if the hemocytes contributed other, non-immune, functions to the insects that produce them. In the following sections, we briefly address the non-immune roles and actions of insect hemocytes.

There is substantial literature on the biological roles of insect hemocytes based on research with *D. melanogaster*. Because some of these actions are basic to insect biology, in this essay we put forth relatively new information on *Drosophila* hemocytes as likely representing hemocyte actions in insects generally. Here, we address the sources of *Drosophila* hemocytes, which occur in two lineages [18,19]. The embryonic/resident lineage begins in the embryonic stage mesoderm in the developing head. These are self-renewing cells that differentiate into plasmatocytes and migrate through the embryo. They settle in organs and tissues within larvae and proliferate. The *Drosophila* lymph glands (LG) are hematopoietic organs that produce the LG lineage. The LG cells proliferate in the embryonic stage, and they begin to differentiate into plasmatocytes in mid-second instar larvae.

## 2. Hemocyte Actions during Molting

Wigglesworth studied insect physiology using the blood-sucking bug, *Rhodnius prolixus*, as a model insect. He reported on hemocyte actions during molting in *R. prolixus*. Briefly, he found that separate injections of trypan blue, india ink, or iron saccarate (a form of dietary iron used to treat people with iron deficiency anemia) into fourth-instar larvae resulted in what he called ‘blocking’ the hemocytes, which led to delayed molting [20]. He also reported that hemocytes contribute to the formation of connective tissue in *R. prolixus* [21]. During this period, many plasmatocytes accumulate on the inner surface of the basement membrane, where they spread rod-like inclusions onto the membrane surface. It was thought the inclusions were the precursors of the connective tissue. In his follow-up paper on staining results, he concluded the hemocytes contribute to thickening the basement membrane and also to thickening muscle insertions [21]. We surmise hemocytes act in constructing insect body structures during molting.

Hemocyte population densities undergo large swings, with substantial increases during molting and declines afterwards. One mechanism of rapidly increasing or decreasing the hemocyte population is to attach or detach cells from inner body walls or organ surfaces as appropriate to the physiological situation. Selectins are cell surface carbohydrate chains that mediate the selective adhesion of mammalian cells to other cells or tissues [22]. Okazaki et al. [23] reported that the changing hemocyte densities in silkworms, *Bombyx mori*, during molting are regulated by attachment and separation of hemocytes, also via selectins. The authors injected enzymes that cut carbohydrate chains, such as glycoside hydrolase and neuraminidase, into silkworms and found the treatments led to increases in circulating hemocyte densities. The authors concluded that insect selectins feature sugar chains that operate in mechanisms analogous to mammalian selectins to rapidly regulate circulating hemocyte populations.

Hemocytes produce and secrete various proteins and peptides. In their study of the sources of peptides in the larval integument of the larger canna leafroller, *Calpodes ethlius*, Sass et al. [24] found some peptides were secreted from the epidermis into the hemolymph and the cuticle that were presumed to be transepidermal. The authors identified four classes of hemocytes in their leafroller: granulocytes, oenocytoids, plasmatocytes, and spherulocytes. They used antibodies to identify the source of one peptide, which they called T66, that occurs in large abundances in the cuticle and in the hemolymph. Testing each hemocyte type against antibodies to T66, they found T66 only occurs in spherulocytes, which they presumed to be the source of it. These peptides are essential to development because they make up part of the epidermis. Sass et al. [24] concluded that integument peptides are sourced not just from the epidermis but also from hemocytes. We surmise hemocytes are essential actors in building insect body structures. Generally, cuticular proteins are synthesized by the epidermis, not hemocytes. We note the Sass group did not record the synthesis of cuticular proteins in hemocytes from larval tobacco hornworms, *Manduca sexta* [25].

## 3. Hemocytes Contribute to Surviving Severe Hypoxia

While the Earth is surrounded by an oxygenated atmosphere, hypoxia is a common environmental constraint in plant and animal biology. Within animals, most malignant tumors are exposed to hypoxia because they grow faster than their oxygen supply. The tumors respond with complex signaling that can lead to tumor metastasis [26]. Aquatic environments are hypoxic relative to terrestrial ones. Some aquatic environments are severely hypoxic due to the serious environmental problem of eutrophication, a process of enriching aquatic ecosystems with nutrients, often agricultural nutrients such as phosphorus, that lead to algal blooms. The algal blooms reduce oxygen concentrations and can lead to an increased frequency of anoxic periods and fish kills [27]. One of the most challenging areas is the coastal ocean on the northern Gulf of Mexico continental shelf, next to the Mississippi River. Currently, the hypoxic areas cover about 23,000 km^2^. Beyond fish kills, local ecosystems, people, and economies are at risk [28].

Some insects, such as dung beetles, are routinely exposed to severe hypoxia and hypercapnia in dung, where oxygen concentrations can go down to 1–2% [29]. Whipple et al. [29] investigated how long five dung beetle species could survive immersion in hypoxic water, using the time to 50% mortality (LT_50_) as their endpoint. The authors created hypoxic water by bubbling nitrogen through the water. They recorded LT_50_ values from five dung beetle species, ranging from 7 to 37 h for *Aphorism haemorrhoidalis* [30]. Overall, animals and plants can be exposed to hypoxia in a wide range of environments. While hemocytes contribute to surviving hypoxia, many insects die in water and other hypoxic environments. For example, Cavallaro et al. [31] reported that many carrion beetles drowned overnight in pitfall traps that became flooded. The authors determined the rapid deaths were due to enhanced microbial activity and carbon dioxide production, which rapidly depleted oxygen from the water.

Azad et al. [32] investigated the influence of hypoxia treatments in animals, using *D. melanogaster* as their animal model. They reported that hypoxia led to substantial changes in gene expression in adult flies. They devised two hypoxia treatments, intermittent (IH) and constant (CH) hypoxia. The 2.5 h CH treatments led to up- and down-regulation of a wide range of gene families. These include GO terms, to name a few of them, ‘response to unfolded protein’, ‘proteolysis’, ‘chitin metabolic process’, ‘response to stress’, and ‘lipid metabolic process’. IH treatments led to changes in the ‘antibacterial humoral response’, among others. Among categories, they found ‘response to unfolded proteins’ had the highest increase in gene expression. They reported that genes encoding heat shock proteins (HSP) 23, 68, and 70 were significantly overexpressed. The authors went on to discover that expression of a gene encoding HSP70 was substantially up-regulated, not in muscles or the heart, but in hemocytes [33]. The hemocyte protein conveys to the flies a high level of survival during exposure to severe hypoxia for days. It also led to increased survival from induced oxidation, caused by feeding them the strong oxidant paraquat. Thus, hemocytes produce a protein that boosts the survival of severe challenges.

## 4. PPO Actions beyond Immunity

Insect hemocytes produce and release PPO, one of the many type-3 copper proteins, into the hemolymph circulation. The inactive zymogen is activated via a number of mechanisms that vary among species. The active PO is a major actor in humoral and cellular immunity. Shrestha et al. [34] reported that in the lepidopteran, *Spodoptera exigua*, PPO is released from a specific hemocyte type called oenocytoid. The release action is due to a PGE_2_ receptor that triggers oenocytoid lysis. In their review, Lu et al. [35] surfaced several non-immune PPO actions. PPO is released from the second hindgut section of the silkworm, *Bombyx mori* [36]. The active hindgut PO melanizes the green frass into black frass. The authors isolated entire alimentary canals from larvae and showed that the chewed-up mulberry leaves remained green through three sections of the midgut and the first of two hindgut sections. In the second hindgut section, the mulberry frass becomes black. They made two points. First, the melanization step is due to active PO and not to laccase or peroxidase, and second, the PO came from hindgut cells and not from the hemolymph. Finally, they showed the hindgut PPO is responsible for regulating the numbers of bacteria in the hindgut and frass. This may be a form of prophylactic immunity because melanization operates to prevent the bacteria-laden frass from contaminating areas near the silkworms. An et al. [37] described another form of prophylactic immunity during molting in which the central nervous system directs the biosynthesis of AMPs during the vulnerable period after a molt when a new cuticle is not yet hardened to the extent that it becomes a barrier to infections.

While there is extensive literature on the topic, we look forward to continuing active research, particularly research toward developing knowledge on PPO and PO into new pest management tools.

## 5. Hemocytes Produce Vitellogenin in a Rice Leafhopper

Some plant viruses seem to manipulate their host plants and their insect vectors in a way that facilitates viral transmission among host plants. For one example, the rice dwarf virus (RDV) alters the green rice leafhopper, *Nephotettix cincticeps*, and host preferences. Wang et al. [38] reported that RDV-infected leafhoppers preferred RDV-free rice plants over RDV-infected plants, and RDV-free leafhoppers selected RDV-infected rice plants over RDV-free plants. The insect selection was based on plant odors, as infected plants emitted different odors from the RDV-free plants. The authors inferred RDV influences the behaviors of its vector by changing host plant physiology.

Huo et al. [38] found that the rice stripe virus (RSV) also manipulates its vector insect, the small brown planthopper, *Laodelphax striatellus*, to facilitate its transmission. Generally, *L. striatellus* acquires the virus from infected rice plants. Once in its vector insect, the virus infects a range of tissues, including salivary glands. When an infected vector feeds on a non-infected host plant, it transmits the virus to the plant. The virus also invades the ovaries, from whence it is vertically transmitted to offspring. In this planthopper species, the hemocytes produce substantial amounts of vitellogenin (Vg), an unusual adaptation because Vg is generally synthesized in insect fat bodies. The authors discovered that *L. striatellus* Vg is processed in different ways among tissues and that only the hemocyte-produced Vg binds RSV. This is due to a Vg subunit that is present only in the hemocyte-produced Vg. While Vg is generally produced in females, *L. striatellus* nymphs and males also synthesize Vg solely in hemocytes. The hemocyte-produced Vg binds the RSV.

The authors used RNA interference to knock down the synthesis of the hemocyte-produced Vg. They injected RSV ribonucleoprotein particles directly into the vector hemocoel, and at selected times after injection, RSV titers were assessed by qRT-PCR. They detected significantly reduced RSV titers in hemolymph and salivary glands, but not in the midgut or fat body. Huo et al. [39] concluded that hemocyte-produced Vg facilitated virus transmission by protecting RSV during hemolymph and salivary gland transport. Here, we appreciate the role of *L. striatellus* hemocytes in facilitating RSV transmission among rice plants by producing an unusual form of Vg. Hemocyte delivery of pathogens (Figure 1) is also known in mosquitoes that transmit arboviruses due to their relatively high susceptibility to viral infection because they lack basal lamina [40]. The infected hemocytes circulate to disseminate the viral pathogens to other tissues, such as the salivary glands. In *Aedes aegypti*, specific galactose-specific C-type lectins (mosGCTL-1 and mosGCTL-3) are highly expressed in hemocytes and facilitate WNV and DENV2 infections by binding to the viral protein E to facilitate viral entry into multiple mosquito tissues [41,42]. The lectins also mediate anti-viral responses by recognizing viral coat proteins and activating PPO and phagocytosis. These opposite uses of identical lectins by the host and pathogen suggest the complexity of their molecular interactions in hemocytes.

## 6. *Drosophila* Hemocytes Act in Cancer Recognition and Responses

Hundreds of papers based on studies with *D. melanogaster* reporting progress in cancer research have appeared since 2000 [43]. Research into mammalian and *Drosophila* cancer defense mechanisms involves immune responses to the presence of tumor cells. Although this essay addresses hemocyte actions that lie beyond immunity, we include a section on hemocyte reactions to the presence of tumor cells in *Drosophila* larvae. The point is to separate hemocyte immune reactions to invasions and infections from hemocyte reactions to cells natively present in the body that are somehow altered in the process of becoming tumor cells.

In fruit fly development, the scribble (*scrib*) genes, lethal (2) giant larvae (*l(2)gl*), and disc large 1 (*dlg*) act in establishing the apical-basal polarity of epithelial cells. They also encode tumor suppressors [44]. Parisi et al. [45] used several fruit fly strains, some with loss-of-function alleles, to document hemocyte-mediated two-way communication between imaginal tumors and the fat body. Contact with tumors leads to the release of the ligand, Spätzle, from hemocytes. Spätzle activates the *Drosophila* fat body tumor necrosis factor, Eiger, which leads to tumor cell death via apoptosis.

Araki et al. [46] used the *Drosophila* hemocyte tumor mutant line, the multi-sex combs (mxc^mbn1^) mutants, to investigate another response to tumor cells. These larvae suffer malignant hyperplasia in their LGs, the larval hematopoietic organ. Hyperplasia is not cancer; the term represents an abnormal accumulation of cells that will likely become cancer. The authors document several points about these mutants. The LGs in these mutants are about 5× larger than LGs in normal larvae. They also showed that fat body genes encoding 6 AMPs, such as drosomycin, defensin, diptericin, metchnikowin, attacin A, and cecropin A2, were highly expressed in the mxc^mbn1^ mutants compared to controls. Importantly, they showed that these AMPs suppressed LG tumor growth. Parvy et al. [47] investigated the mechanism of how a specific AMP, Defensin, kills tumor cells. They deleted the gene encoding Defensin, after which they recorded fewer tumor cell deaths and larger tumors. On closer examination, they noted that the Defensin protein interacted with tumor cell membranes. The mammalian tumor necrosis factor, TNF, exerts pleiotropic effects on tumors. The *Drosophila* TNF homolog, Eiger, also drives tumor cell death.

Cell membranes are typically imagined as a bilayer composed of two leaflets. The head groups are located on the outside of the membrane, and the fatty acid chains are on the inside. The phospholipid, phosphatidylcholine (PC), with a positively charged head group, is usually on the outer leaflet, and phosphatidylserine (PS), with a negatively charged head group, is typically on the inner leaflet facing the cytoplasm. In *Drosophila* tumor cells, PS is transposed to the outer leaflet, where the negatively charged head group is available to the positively charged Defensin, which leads to tumor cell death. Hence, the altered structures of biomembranes in tumor cells are one molecular mechanism of Defensin-driven cell death [47].

As just mentioned, there are many papers on *Drosophila* tumors of tangible value in human cancer biology research. Our point in this section is that insect hemocytes are major actors in insect cancer biology.

## 7. Clearing Apoptotic Cells from CNS

The differentiated plasmatocytes exert important actions in several tissues. In the Central Nervous System (CNS), plasmatocytes operate to clear apoptotic cells during *Drosophila* development and adulthood. Developmental processes involve a great deal of apoptosis. Differentiated hemocytes, or plasmatocytes, are responsible for clearing apoptotic cells from embryos. These cells express a gene encoding the apoptotic cell-clearing receptor, Six-microns-under (Simu). Loss of this gene leads to pathologically high levels of apoptotic cells because they are not cleared from tissues. Simu also acts to clear necrotic materials. Roddie et al. [48] interpreted these results to show that Simu is a detector and clearer of the damaged self. We note that while *Drosophila* hemocytes clear damaged and apoptotic cells and material from embryos, they are not the sole cell clearers. Glia cells (often denoted ‘glia’) are non-neuronal cells that also clear apoptotic cells and operate in providing gas exchange and nutrients to the CNS [49]. More to the point, during *Drosophila* CNS developmental remodeling, many neurites are destined to be pruned and cleared. Hemocytes also act in clearing the pruned neurites, which is necessary to prevent inflammation and maintain homeostasis. Although hemocytes act in the process, Han et al. [50] proposed that epidermal cells, not hemocytes, are the primary phagocytes for pruning neurons. We conclude that clearing apoptotic cells and pruned neurites are essential developmental processes that require several cell types, including hemocytes.

## 8. Hemocytes Create Hematopoietic Niches

Differentiated cell populations are generally produced from stem cells. Importantly, populations of stem cells and differentiated cells are maintained in a balance that is necessary for lifelong tissue upkeep. Stem cells mostly divide asymmetrically, giving rise to another stem cell and a second cell destined to differentiate [51]. Stem cells generally reside in micro-compartments called ‘niches’ that are supportive and protective [51,52]. Hematopoietic stem cells (HSCs) differentiate into blood cells in vertebrates and into hemocytes in insects [52]. In *Drosophila*, hemocytes appear in early embryos. *Drosophila* larvae develop hematopoietic pockets found in lateral and dorsal patches along the length of the body, and these pockets support sessile hemocytes. These pockets occur in the gaps between the epidermis and the body wall muscles. Hemocytes remain connected to the basement membranes, which is necessary to complete the formation of sessile hematopoietic spaces. Csordás et al. Additionally, [52] reported on a specific receptor-ligand interaction that keeps sessile hematocytes connected to the body wall. On hemocytes, the phagocytosis receptor, Eater, connects to a collagen ortholog, Multiplexin, in the epidermal basement membrane. The connections create the hematopoietic niches in *Drosophila*.

## 9. Hemocytes Produce and Transport the Lipoprotein, ApoLP-III

The insect lipoprotein ApoLP-III is synthesized in the fat body and hemocytes. The most well-studied biological role of ApoLP-III is transporting lipids, in the form of diacylglyerols, in aqueous hemolymph from the fat body to metabolizing tissues such as flight muscles. The lipoproteins also act in insect immunity. Kim et al. [53] reported that hemocytes in the fall webworm, *Hyphantria cunea*, biosynthesize ApoLP-III as well as take it up from hemolymph. They found that injecting ApoLP-III with the bacterium *Escherichia coli* led to substantial increases in the expression of genes encoding AMPs. In this brief section, we make the point that, aside from their direct actions in immunity, hemocytes also produce and release a lipoprotein that operates in lipid transport and immunity.

## 10. Hemocytes Transport Iron

Iron is essential in many biological processes, such as DNA and RNA synthesis, cellular respiration, and xenobiotic metabolism. Paradoxically, an overabundance of iron can disrupt cellular activities. Most iron in mammals is associated with hemoglobin in red blood cells. As insects do not express red blood cells, their needs and transport of iron differ fundamentally from mammals. Insects and mammals produce iron-transporting proteins called ferritins, although insect ferritins differ from their mammalian counterparts. Drawing on Locke and Nichol [54], insect ferritin from the Brazilian skipper, *Calpodes ethlius*, is composed of two subunits of differing sizes, 24 and 31 kd, and two smaller subunits of 26 and 28 kDa. The complete molecule is very large, over 600 kDa. The cellular locations of the iron-loaded holoprotein differ among species, although it occurs in the vacuolar system of some species. In particular, *C. ethlius* apoferritin occurs in hemocytes. Locke [55] found apoferritin in hemocyte preparations from *C. ethlius*, resolved on transmission electron microscopy. He loaded normal hemolymph with iron and observed particles on the rough endoplasmic reticulum of hemocytes, which he interpreted as iron-loaded holo-ferritin and suggested hemocytes could be a source of hemolymph ferritin.

Several ferritin chains have been cloned from insects. For a specific example, Qia et al. [56] cloned a heavy chain ferritin fragment from the Colorado potato beetle. The protein features 213 amino acids, a 19 amino acid signal peptide, and a seven-amino acid ferroxidase center. We do not recite all of them here, other than to say information on insect ferritins is rapidly increasing. Pham et al. [57] cloned a subunit of the tobacco hornworm, *Manduca sexta*, ferritin. They found the mRNA encoding the subunit is expressed in the midgut, fat body, and hemocytes.

Later, Pham, and Winzerling [58] compared insect and mammalian ferritins. They found that, unlike mammalian ferritin, the insect proteins have more mass, have glycosylated subunits, and occur in tetrahedral symmetry with 12 heavy chain and 12 light chain subunits. The insect ferritins occur in the vacuolar system of hemocytes, where they serve in iron transport. The mammalian ferritins occur in octahedral symmetry with 24 heavy and 24 light chain subunits. The mammalian ferritins occur in the cytoplasm, where they serve in iron storage rather than transport.

Iron is an essential nutritional element and, at the same time, a dangerously reactive element that can lead to serious oxidative damage. Gonzalez-Morales et al. [59] noted that there are no reports of mRNAs encoding ferritin in insect hemocytes and that ferritin accumulates mainly in the midgut and hemocytes of *Drosophila* embryos. Pham et al. [57] show mRNA encoding ferritin in hemocytes, the fat body, and the midgut of *M. sexta*, from which we generalize that insect hemocytes produce ferritin and transport iron throughout the body.

## 11. Conclusions

Research into the biology of insect hemocytes already has a lengthy history, documented in this essay (Table 1) with some of the original publications. Over the years, we have seen scientists apply new tools to hemocyte research as they become available. For a single example, several authors proposed various hemocyte classifications based on careful morphological studies. More recently, scientists have applied molecular tools, such as single-cell RNA sequencing (scRNAseq), to achieve a more detailed appreciation of hemocyte types and actions. This work is not restricted to insects, as recent papers also focus on other invertebrates. For a single example, Pichon et al. [60] applied scRNAseq analysis of hemocytes from the snail, *Biomphalaria glabrate*, an intermediate host of *Schistosoma mansoni*. They identified seven transcriptomic populations within the hemocytes, an improvement on the two to five hemocyte populations based on morphological studies. Similarly, Li et al. [61] created the Fly Cell Atlas, which facilitated, among other gains, the identification of cell type-specific markers across the entire animal. The work extended to a very large scale, as many authors sequenced 570,000 cells. Biotechnology and biological science will continue to advance, and we foresee continuing discovery in the science of insect- and invertebrate-hemocytes. Drawing on Dhahbvi et al. [62], who modelled the sterile insect technique, we also look forward to the applications of additional advanced tools, including mathematical modeling, to understand interactions of hemocytes with infectious microbes at population levels. In our view, the coming years of research into invertebrate hemocytes will lead to valuable new knowledge and effective new tools for application in insect pest management programs.

## Figures and Tables

**Figure 1 cells-12-00599-f001:**
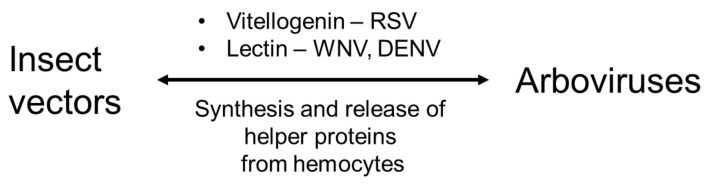
Unique hemocyte function in insect vectors for transmission of arboviruses. Hemocytes produce helper proteins to deliver the arboviruses to the salivary gland in *L. striatellus* for rice stripe virus (RSV) and *A. albopictus* for West Nile virus (WNV) and Dengue virus (DENV).

**Table 1 cells-12-00599-t001:** Physiological functions of hemocytes in immune and non-immune processes in insects.

Category	Physiological Functions	Model Insects	References
Immunity	Cellular immunity	*Manduca sexta*	[2]
	Phagocytosis		
	Nodulation		
	Encapsulation		
	Humoral immunity	*Drosophila melanogaster*	[6]
	Antimicrobial peptides		
	Melanization		
Non-Immunity	Molting and development	*Rhodnius prolixus*	[20,21]
	Surviving severe hypoxia	*Drosophila melanogaster*	[33]
	Phenoloxidase production	*Bombyx mori*	[36]
	Vitellogenin production	*Laodelphax striatellus*	[38,39]
	Recognition of cancer cells	*Drosophila melanogaster*	[45,46,47]
	Clearing apoptotic cells	*Drosophila melanogaster*	[48,49]
	Hematopoiesis	*Drosophila melanogaster*	[51,52]
	Lipoprotein synthesis	*Hyphantria cunea*	[53]
	Iron transport	*Calpodes ethlius*	[55]

## Data Availability

Not applicable.

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
