# Peer review of "Beyond Cellular Immunity: On the Biological Significance of Insect Hemocytes"

_cells, 2023, doi:10.3390/cells12040599_

Round 1

Reviewer 1 Report

This is a review manuscript regarding the physiological roles of hemocytes beyond the cellular immunity in insects, which include molting and development, surviving severe hypoxia, producing PO precursor/vitellogenin, recognition and responses to cancer, non-immune actions and clearing apoptotic cells in the CNS, developing hematopoietic niches, synthesis and transport of lipoprotein, and hemocyte roles in iron transport. The described and summarized story will benefit and contribute to the study of insect immune systems. This is a well-written manuscript, I couldn’t find any major issues in the manuscript, and it can be published after a few editorial edits.

Line 98-100: I think the sentence is unnecessary, although Wigglesworth is one of the pioneers in insect physiology. It could replace with “Wigglesworth studied insect physiology using the blood-sucking bug, Rhodnius prolixus, as a model insect.”

 Line 284: central nervous system (CNS) 

Author Response

his is a review manuscript regarding the physiological roles of hemocytes beyond the cellular immunity in insects, which include molting and development, surviving severe hypoxia, producing PO precursor/vitellogenin, recognition and responses to cancer, non-immune actions and clearing apoptotic cells in the CNS, developing hematopoietic niches, synthesis and transport of lipoprotein, and hemocyte roles in iron transport. The described and summarized story will benefit and contribute to the study of insect immune systems. This is a well-written manuscript, I couldn’t find any major issues in the manuscript, and it can be published after a few editorial edits.

Comment #1-1Line 98-100: I think the sentence is unnecessary, although Wigglesworth is one of the pioneers in insect physiology. It could replace with “Wigglesworth studied insect physiology using the blood-sucking bug, Rhodnius prolixus, as a model insect.”

ResponseCorrected as suggested

Comment #1-2 Line 284: central nervous system (CNS) 

ResponseCorrected as suggested

Reviewer 2 Report

I like the idea of presenting the function of insect hemocytes other than immunity. The review article is interesting and in my opinion is of interest of broad audience. I have only three comments:

1.       The authors should mention also the function of sferulocytes in supply of cuticle components in Lepidoptera.

2.       Also the role of hemocytes in stress response is missing, like synthesis of JAK/STAT ligand

3.       At the end of the article I miss a scheme with main conclussions concerning the role of insect hemocytes in insect physiology other than immunity.

Author Response

I like the idea of presenting the function of insect hemocytes other than immunity. The review article is interesting and in my opinion is of interest of broad audience. I have only three comments:

Comment #2-1: The authors should mention also the function of spherulocytes in supply of cuticle components in Lepidoptera.

ResponseAgree, we cited Sass et al [24] and briefly discussed the point in lines 124-138. Now in the revision, we added another citation [25] from the Sass group to balance the statement with their later work showing M. sexta larvae do NOT source cuticular proteins from hemocytes.  

Comment #2-2: Also the role of hemocytes in stress response is missing, like synthesis of JAK/STAT ligand

Response: Agree, we added a new section to consider hemocyte actions in stress responses.  

Comment #2-3: At the end of the article I miss a scheme with main conclusions concerning the role of insect hemocytes in insect physiology other than immunity.

ResponseAgree, we developed a new figure to show our main points. 

Reviewer 3 Report

This is an interesting study; the authors have collected the relevant information. The manuscript is generally well-written and structured. However, in my opinion, the manuscript has some shortcomings regarding logical discussions and the flow of information in the text. Below I have provided a few remarks to further improve the manuscript's quality. Given these shortcomings, the manuscript requires major revisions.

1.       I would like to suggest the addition of the most recent literature, as the literature cited is too old. Most, if not all, literature should be from the last 4-6 years.

2.       There are no figures or tables in the whole review, and there should be at least a few figures in this review to summarize findings or highlight new ideas.  

3.       There is a lack of logical discussion in almost each review section, and it seems like a mere literature review. In several instances, the authors only cited what other studies resulted in without any logical discussion. This aspect should be improved in the whole manuscript. 

4.       There should be a section on future prospects at the end of the manuscript. 

Author Response

This is an interesting study; the authors have collected the relevant information. The manuscript is generally well-written and structured. However, in my opinion, the manuscript has some shortcomings regarding logical discussions and the flow of information in the text. Below I have provided a few remarks to further improve the manuscript's quality. Given these shortcomings, the manuscript requires major revisions.

Comment #3-1: I would like to suggest the addition of the most recent literature, as the literature cited is too old. Most, if not all, literature should be from the last 4-6 years.

Response: We take the point and cited some of the newer publications. However, our use of some older publications is not an accident. In this essay, these older publications are meant to convey the point that there is a lengthy history of research into insect hemocytes, some of which has contributed fundamental and long-lasting knowledge on the biology of insects and their hemocytes. In our view, this is an important consideration in our current era of rapid review/revision/publication cycles. 

Comment #3-2: There are no figures or tables in the whole review, and there should be at least a few figures in this review to summarize findings or highlight new ideas.  

ResponseAgree, we understand your point and added a figure to summarize the overall points of the manuscript. 

Comment #3-3: There is a lack of logical discussion in almost each review section, and it seems like a mere literature review. In several instances, the authors only cited what other studies resulted in without any logical discussion. This aspect should be improved in the whole manuscript. 

Response: We take your point. We added more discussion to selected sections, although not all of them. We note that section 2 on hypoxia begins with a broad treatment of hypoxia, meant to discuss the global and local issues in the topic and we feel further discussion is necessary. Similarly, we did not add more discussion to section 6 on cancer.     

Comment #3-4: There should be a section on future prospects at the end of the manuscript.

ResponseAgree, we added a new final paragraph to briefly discuss the point. 

Round 2

Reviewer 3 Report

Although the authors did not add much detail as requested. However, the manuscript can be accepted in its current form. 

Author Response

We replaced the figure with Table to be clear.